# Transframer: Arbitrary Frame Prediction with Generative Models

## Abstract

We present a general-purpose framework for image modelling and vision tasks based on probabilistic frame prediction. Our approach unifies a broad range of tasks, from image segmentation, to novel view synthesis and video interpolation. We pair this framework with an architecture we term Transframer, which uses U-Net and Transformer components to condition on annotated context frames, and outputs sequences of sparse, compressed image features. Transframer is the state-of-the-art on a variety of video generation benchmarks, is competitive with the strongest models on few-shot view synthesis, and can generate coherent 30 second videos from a single image without any explicit geometric information. A single generalist Transframer simultaneously produces promising results on 8 tasks, including semantic segmentation, image classification and optical flow prediction with no task-specific architectural components, demonstrating that multi-task computer vision can be tackled using probabilistic image models. Our approach can in principle be applied to a wide range of applications that require learning the conditional structure of annotated image-formatted data.

## 1 Introduction

We introduce a framework that unifies a wide range of tasks under the umbrella of conditional frame prediction, and demonstrate its efficacy on an extensive range of challenging image modelling and computer vision tasks.

Our proposed model, which we call *Transframer*, is trained to predict the probability of arbitrary frames conditioned on one or more annotated context frames. Context frames could be previous video frames, along with time annotations, or views of a scene with associated camera annotations. Transframer is a likelihood-based autoregressive method analogous to WaveNet (Oord et al., 2016) and large-scale language models (Brown et al., 2020), implemented with Transformers (Vaswani et al., 2017) and U-Nets (Ronneberger et al., 2015), whose underlying representation of the data uses the sparse discrete cosine transform (DCT) image representation introduced by Nash et al. (Nash et al., 2021).

Unlike deterministic methods for frame-to-frame modelling (Eigen & Fergus, 2015; Kokkinos, 2017), Transframer maps context frames to a probability distribution, which allows uncertainty to be quantified and diverse outputs to be obtained through sampling. While autoregressive models are typically expensive or prohibitive to evaluate and sample from, Transframer's sparse representations allow more efficient computation than raw pixels, which we demonstrate by sampling coherent 30 second-long videos.

We applied Transframer to a wide range of tasks, ranging from video modelling, novel view synthesis, semantic segmentation, object recognition, depth estimation, optical flow prediction, etc., and found promising results in all cases.

Our work makes the following key contributions:

- Transframer's quantitative performance is state-of-the-art on video modelling and competitive on novel view synthesis.

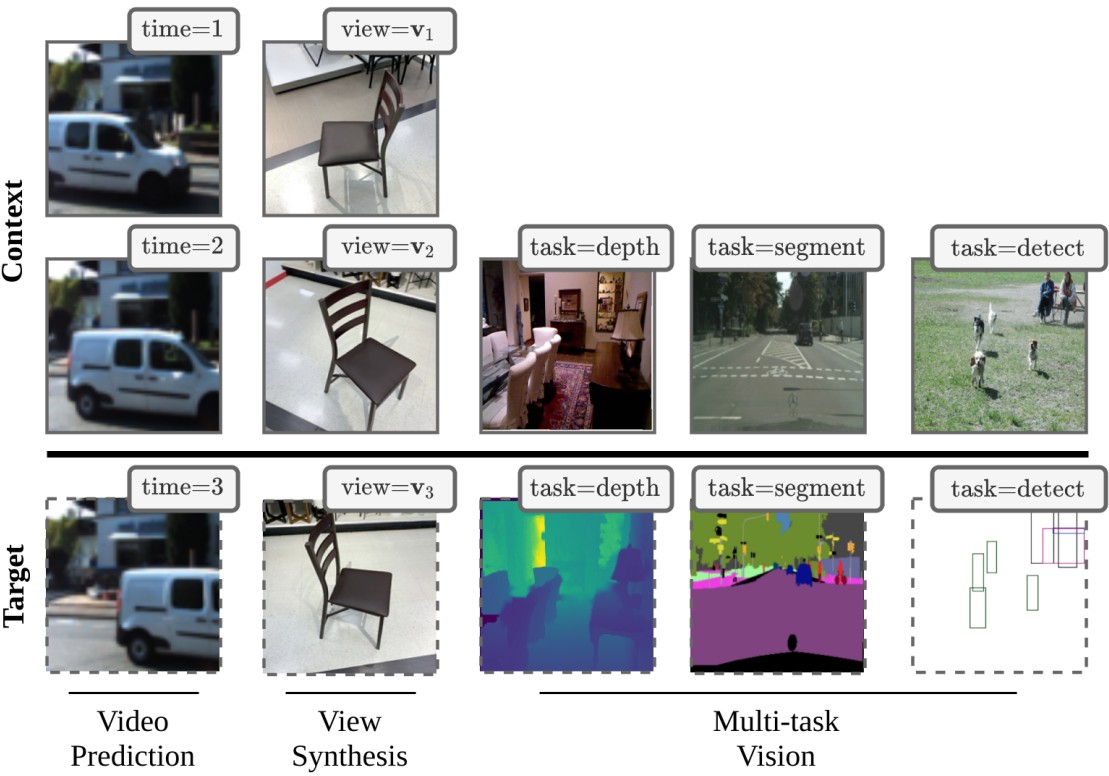

Figure 1: A framework for general visual prediction. Given a collection of context images with associated annotations (time-stamps, camera viewpoints, etc. ), and a query annotation, the task is to predict a probability distribution over the target image. This framework supports a range of visual prediction tasks, including video modelling, novel view synthesis, and multi-task vision.

- We introduce residual sparse DCT representations as a basis for generative video modelling.
- Transframer is general-purpose and can be trained on a wider variety of modelling tasks than are traditionally tackled by more domain-specific methods.
- We show how to generate long video sequences (30 seconds) using our probabilistic framework.

## 2 Background

To contextualize this work, we first provide a description of the prior work we build on, focusing on generative models of compressed image representations, and the DCTransformer generative image model (Nash et al., 2021). An additional discussion of related work on generative video modelling, view synthesis, and alternative frameworks for image-to-image translation can be found in the Appendix A.

**Generative modelling with compressed image representations** To deal with the high dimensionality of image, audio and video data, a number of generative models operate on compressed data representations (van den Oord et al., 2017; Esser et al., 2021; Nash et al., 2021). This is particularly important for likelihood-based generative models, which can otherwise waste their capacity on perceptually unimportant details (van den Oord et al., 2017). For images $\mathbf{I}$ the general approach is as follows:

1. Convert images to compressed codes using an encoder $\mathbf{I}^{\mathrm{code}} = \mathrm{Encoder}(\mathbf{I})$,
2. Train a generative model $p(\mathbf{I}^{\mathrm{code}})$ on the image codes,
3. Decode samples $\mathbf{I}_s^{\mathrm{code}} \sim p(\mathbf{I}^{\mathrm{code}})$ from the model back to images $\hat{\mathbf{I}}_s = \mathrm{Decoder}(\mathbf{I}_s^{\mathrm{code}})$.

VQ-VAE (van den Oord et al., 2017), the most well-known of these approaches, uses a vector-quantized autoencoder network to spatially downsample and compress image data, reducing the size of the image representation while maintaining the most important perceptual details. The resulting image codes are modelled post-hoc using a PixelCNN-style autoregressive model, resulting in samples that are more globally coherent than samples from models trained on raw pixel data. VQ-GAN (Esser et al., 2021) uses an adversarially trained decoder to achieve higher compression rates and image reconstructions with improved perceptual quality. Transformer-based autoregressive architectures are an effective alternatives to PixelCNN, forming the basis of prior generative models in a number of works (Esser et al., 2021; Wu et al., 2021b).

**Sparse DCT image representations** In this work we use the DCT-based image representation proposed by Nash et al. (Nash et al., 2021), rather than raw pixels. The DCT-based image representation is heavily inspired by the JPEG codec (Wallace, 1991); and it encodes data in several stages:

1. The DCT is applied to non-overlapping image patches for each channel in the YUV color space. The UV colour components are optionally 2x downsampled before applying the block-DCT.
2. The resulting DCT data are lossily quantized using a quality-parameterized quantization matrix. Lower quality settings result in stronger quantization, and a greater loss of information.
3. The quantized DCT data are converted to a sparse representation: a list consisting of tuples $\mathbf{d} = (c, p, v)$ of DCT channels, positions, and values for the non-zero elements.
4. The list is sorted for the purposes of generative modelling. As in Nash et al. we order from low-to-high frequency DCT channels, interleaving luminance and color channels. Within DCT channels, list elements are sorted by spatial position in a raster-scan order.

The resulting image representation has a number of key properties. It consists of variable-length sequences $[\mathbf{d}_l]_{l=1}^L$, where the length $L$ depends on the content of the input image. Images with more higher-frequency content result in longer sequences. This contrasts with the compressed representations of VQ-VAE and VQ-GAN, which are fixed size regardless of image content. In addition, the size of the representation and quality of image reconstructions is controllable via the quantization quality setting. We can choose low-quality settings for more efficient training and inference, or high-quality settings for increased image quality.

**DCTransformer** To model the above data representation, Nash et al. (Nash et al., 2021) proposed the DCTransformer architecture. DCTransformer is an autoregressive model that predicts elements of sparse DCT sequences conditioned on all prior elements:

$$p([\mathbf{d}_l]_l) = \prod_l p(\mathbf{d}_l \mid \mathbf{d}_{<l}) \tag{1}$$

$$p(\mathbf{d}_l \mid \mathbf{d}_{<l}) = p(v_l|c_l, p_l, \mathbf{d}_{<l})p(p_l|c_l, \mathbf{d}_{<l})p(c_l|\mathbf{d}_{<l}) \tag{2}$$

The architecture has an encoder-decoder structure, where the encoder processes arbitrarily long sequences of DCT data, and the decoder autoregressively predicts fixed-size chunks of DCT data, conditioned on the encoder's outputs.

During training a target chunk $[\mathbf{d}_l]_{l=t}^{t+C}$ of length $C$ is randomly selected from the DCT sequence, and all prior elements in the sequence $[\mathbf{d}_l]_{l=1}^t$ are treated as inputs. The sparse inputs are converted to a dense *DCT image*. DCT images are 3D arrays, where each spatial location corresponds to a pixel block of size $B{\times}B$, and channels correspond to different DCT components. For images of size $H{\times}W{\times}3$ and DCT block size $B$ the corresponding DCT images will therefore be of size $H/B{\times}W/B{\times}3B^2$. Input DCT images are *partially observed* as they consist only of data from the partial DCT input sequence $[\mathbf{d}_l]_{l=1}^t$, rather than the full data sequence. The use of DCT images enables the encoder to process arbitrarily long input sequences, as regardless of the length they are converted to fixed size DCT image arrays. This contrasts with typical sequence-modelling approaches, where memory and computation scales with sequence length.

A Vision-Transformer encoder processes partial DCT images, and a sequence of output embeddings is passed to a causally-masked Transformer decoder. The Transformer decoder embeds and processes target sequences $[\mathbf{d}_l]_{l=t}^{t+C}$, and uses channel, position and value heads to output predictions for the corresponding list elements. The list elements are treated as discrete values, and predicted using a softmax distribution, with an autore-

Table 1: Target, context, and annotation specification for different task types.

| Task type | $\mathbf{a}^{\text{target}}$ | $\mathcal{C}$ | Description |
|---|---|---|---|
| Video modelling | $t_{\text{target}}$ | $\{(\mathbf{I}_t, t)\}_{t=1}^{t_{\text{target}}-1}$ | Query an image at desired time, $t_{\text{target}}$. The $\mathcal{C}$ contains the sequence of $t_{\text{target}}-1$ previous images and associated timestamps, $t$. |
| Video interpolation | $t_{\text{target}}$ | $\{(\mathbf{I}_t, t)\}_{t=t_0}^{t_{\text{target}}-1}$ $\cup \{(\mathbf{I}_T, T)\}$ | Query an image at desired time, $t_{\text{target}}$. The $\mathcal{C}$ contains a sequence of immediately preceding frames starting at time, $t_0 < t_{\text{target}}$, a future frame at time, $T > t_{\text{target}}$, and associated timestamps, $t$. |
| View synthesis | $\mathbf{v}_{\text{target}}$ | $\{(\mathbf{I}_n, \mathbf{v}_n)\}_{n=1}^{N}$ | Query an image from a desired viewpoint, $\mathbf{v}_{\text{target}}$. The $\mathcal{C}$ contains $N$ other images and associated viewpoints, $\mathbf{v}_n$. |
| Multi-format image translation | $f_{\text{target}}$ | $\{(\mathbf{I}_{f_{\text{RGB}}}, f_{\text{RGB}})\}$ | Query a desired image format, $f_{\text{target}}$, e.g., RGB, depth channel, semantic segmentation, etc. The $\mathcal{C}$ contains an RGB image, annotated $f_{\text{RGB}}$. |

gressive cross-entropy training objective. Writing $\theta_c(\cdot)$, $\theta_p(\cdot)$ and $\theta_v(\cdot)$ for the channel, position and value softmax parameters respectively the objective for target chunk $[\mathbf{d}_l]_{l=t}^{t+C}$ is:

$$\mathcal{L} = \sum_{l=t}^{t+C} \log \theta_v(c_l, p_l, \mathbf{d}_{<l}) \cdot \mathbf{v}_l + \log \theta_p(c_l, \mathbf{d}_{<l}) \cdot \mathbf{p}_l + \log \theta_c(\mathbf{d}_{<l}) \cdot \mathbf{c}_l \tag{3}$$

where $\mathbf{v}_l$, $\mathbf{p}_l$ and $\mathbf{c}_l$ are one-hot vectors associated with indices $v_l$, $p_l$ and $c_l$ respectively. In this work we use the same decoder and training objective, but augment the encoder to additionally process annotated context images.

## 3 Methods

Here, we present two main components of our work. (1) A framework for probabilistic vision that generalizes video prediction, few-shot view synthesis, and other computer vision tasks, (2) An architecture and modelling paradigm that implements that framework called Transframer, which builds on the DCTransformer (Nash et al., 2021) generative image model and data representation.

### 3.1 Generalized conditional image modelling

Our probabilistic formulation is as follows: given an annotation $\mathbf{a}^{\text{target}}$, and context information $\mathcal{C}$, the task is to predict a distribution over target image $\mathbf{I}^{\text{target}}$,

$$p\left(\mathbf{I}^{\text{target}} \mid \mathbf{a}^{\text{target}}, \, \mathcal{C}\right), \tag{4}$$

where $\mathcal{C} = \{(\mathbf{I}_n, \mathbf{a}_n)\}_n$ is a set of context images $\mathbf{I}_n$ and associated annotations $\mathbf{a}_n$. An annotation is metadata about the image such as video timestamps, camera coordinates, image channels, etc. This framework can be applied to a wide range of frame-prediction tasks, as long as we can phrase them in terms of $\mathbf{a}^{\text{target}}$ and $\mathcal{C}$. Some of the tasks we consider, such as novel view synthesis and video generation have previously been expressed in this manner (Eslami et al., 2018; Kumar et al., 2018). Other tasks, including semantic segmentation, depth estimation, or optical flow estimation are frequently modeled within the image prediction framework but typically using task-specific losses. Finally, tasks such as object detection and image classification are generally approached using specialized representations and losses, without any image generation components. Our framework enables us to perform all these tasks using a single unified training objective.

In some cases, we want a predictive distribution over multiple target images $\left\{\mathbf{I}_k^{\text{target}}\right\}_k$ conditioned on queries $\left\{\mathbf{a}_k^{\text{target}}\right\}_k$. For example, to predict multiple consecutive frames of video, or to synthesize views at multiple query viewpoints. The simplest approach is to treat the target images as conditionally independent:

$$p\left(\left\{\mathbf{I}_k^{\text{target}}\right\}_k \mid \left\{\mathbf{a}_k^{\text{target}}\right\}_k, \mathcal{C}\right) = \prod_k p\left(\mathbf{I}_k^{\text{target}} \mid \mathbf{a}_k^{\text{target}}, \mathcal{C}\right), \tag{5}$$

This enables parallel generation of the target images, but does not capture any additional dependencies between the target images. In cases where the target images are constrained by the the context, such as view synthesis with a large collection of input views, this is not a significant problem. However, whenever there exist strong dependencies between target images, as in video modelling for example, sampled output images are likely to be inconsistent with one another. To address this issue, we can instead predict target images sequentially, updating the context set with newly generated images at each step. This corresponds to the following autoregressive model:

$$p\left(\left\{\mathbf{I}_k^{\text{target}}\right\}_k \mid \left\{\mathbf{a}_k^{\text{target}}\right\}_k, \mathcal{C}\right) = \prod_k p\left(\mathbf{I}_k^{\text{target}} \mid \mathbf{a}_k^{\text{target}}, \mathcal{C}_k\right) \tag{6}$$

$$\mathcal{C}_k = \{(\mathbf{I}_n, \mathbf{a}_n)\}_n \cup \left\{(\mathbf{I}_p^{\text{target}}, \mathbf{a}_p^{\text{target}})\right\}_{p<k}. \tag{7}$$

Predictions for each target image will take into account all previously generated images, and can therefore capture the required dependencies.

**A unified task formulation** We apply our framework to four different general task specifications, which are distinguished by their target annotations $\mathbf{a}^{\text{target}}$ and context set $\mathcal{C}$, as described in Table 1.

**Training objective** A range of modelling paradigms can be used to estimate the predictive distributions in Equation 4, with DCTransformer with sparse DCT representations being a particular example. Other examples include conditional VAEs such as GQN (Eslami et al., 2018), conditional GANs as in pix2pix (Isola et al., 2017), or conditional diffusion-based models (Ho et al., 2020). The kind of training objective depends on the model class used. In this work, we build on the DCTransformer, and therefore use the corresponding autoregressive training objective described in Section 2.

## 3.2 Transframer architecture

To estimate predictive distributions over target images, we require an expressive generative model that can produce diverse, high-quality outputs. DCTransformer (Section 2, (Nash et al., 2021)) produces compelling results on single image domains, but is not designed to condition on the multi-image context sets $\{(\mathbf{I}_n, \mathbf{a}_n)\}_n$ that we require. As such, we extend DCTransformer (Section 2, (Nash et al., 2021)) to enable image and annotation-conditional predictions. In particular, we replace DCTransformer's Vision-Transformer-style encoder, which operates on single (partially observed) DCT images, with a multi-frame U-Net architecture, that processes a set of annotated context frames along with the partially observed target frame. As in DCTransformer, we pass output embeddings from the encoder to a Transformer-based decoder, and autoregressively predict sparse DCT tokens (Figure 2a).

**Multi-frame U-Net** The input to the U-Net is a sequence consisting of $N$ context DCT frames (See Section 2), and a partially observed target DCT frame. Annotation information is provided in the form of vectors associated with each input frame. Initially, the DCT image channels are linearly projected to an embedding dimensionality $E$ using $1 \times 1$ convolutions. We add horizontal and vertical positional embeddings to the embedded DCT images, and incorporate annotation information $\mathbf{a}$ by adding linearly projected annotation vectors.

The core component of our U-Net is a computational block that first applies a shared NF-ResNet convolutional block (Brock et al., 2021) to each input frame, and then applies a Transformer-style self-attention block to aggregate information across frames (Figure 2b). NF-ResNet blocks consist of grouped convolutions followed by a squeeze-and-excite layer (Hu et al., 2018), and are designed for efficient performance

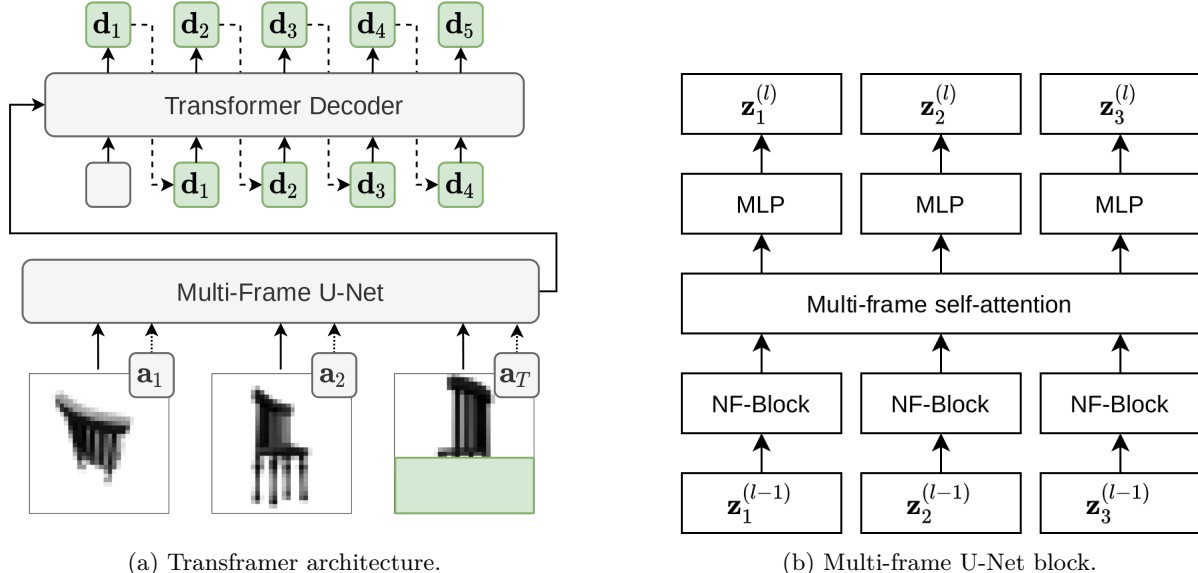

(a) Transframer architecture.

(b) Multi-frame U-Net block.

Figure 2: (a) Transframer takes as input context DCT-images (left and middle), as well as a partially observed DCT-image of the target (right) and additional annotations $\mathbf{a}$. The inputs are processed by a multi-frame U-Net encoder, which operates at a number of spatial resolutions. U-Net outputs are passed to a DCTransformer decoder via cross-attention, which autoregressively generates a sequence of DCT tokens corresponding to the unseen portion of the target image (shown in green). (b) Multi-frame U-Net blocks consist of NF-Net convolutional blocks, multi-frame self-attention blocks - which exchange information across input frames - and a Transformer-style residual MLP. $z_i^l$ is a representation from the $l - th$ block and $i$-th frame.

on TPUs (tpu). The self-attention block is based on a standard Transformer block (Vaswani et al., 2017), with a self-attention layer applied before an MLP. We use one of two self-attention types: In the first case, self-attention is applied to corresponding spatial locations across the frame feature maps. In the latter case, self-attention is applied both spatially and across frames. In some cases, we omit self-attention entirely to reduce memory consumption. We use multi-query attention (Shazeer, 2019), rather than the standard multi-head attention in order to improve sampling speed.

As in a standard U-Net, our multi-frame variant has a downsampling stream, where feature maps are spatially downsampled for more efficient processing, and an upsampling stream where feature maps are spatially upsampled back to the input resolution. Skip-connections are used to propagate information from the downsampling stream to the upsampling stream directly, bypassing low-resolution information bottlenecks. In both streams, multiple convolutional blocks are applied at a given resolution before an upsampling or downsampling block changes the spatial resolution. We use a symmetrical structure with the same number of block at each resolution in both streams.

### 3.3 Residual DCT representations

Consecutive video frames often exhibit substantial temporal redundancy. As an extreme case, in videos with static cameras and backgrounds, pixel data may be almost entirely unchanged across adjacent frames. In the DCTransformer data format (Section 2), a sparse image representation is obtained by storing the positions and values of non-zero DCT coefficients. When working with video data, we can potentially increase the degree of sparsity by storing DCT data only where it changes relative to the previous frame.

Figure 3 visualizes the sparsity of absolute and residual DCT representations for two datasets. The sparsity gains of the residual representation are significant in the static-camera RoboNet dataset (Dasari et al., 2019). However, for the KITTI dataset (Geiger et al., 2012), which features a continuously moving camera,

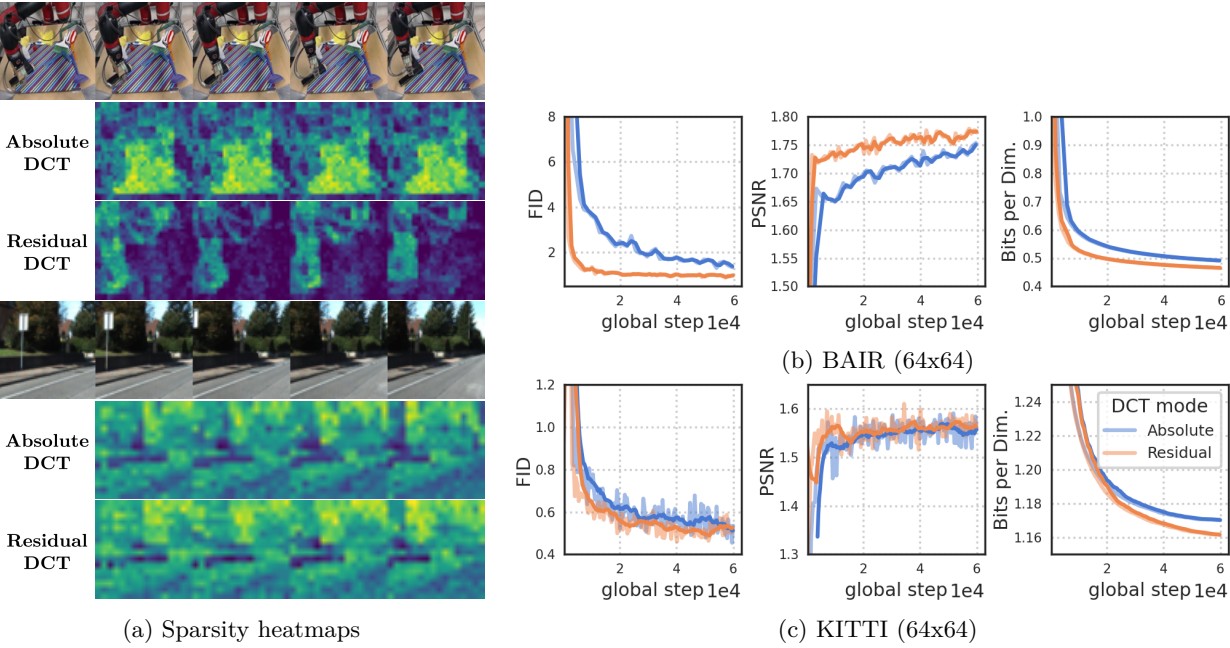

(a) Sparsity heatmaps

(b) BAIR (64x64)

(c) KITTI (64x64)

Figure 3: (a) Comparing the sparsity of absolute and residual DCT representations for RoboNet (128x128; top) and KITTI videos (bottom). RoboNet has static videos with only a few moving elements, so residual frame representations yield a substantial increase in sparsity. KITTI videos typically feature a moving camera, resulting in differences almost everywhere in consecutive frames. In this case the sparsity benefits of residuals is minor. (b) Evaluation metrics for absolute and residual representations on BAIR and KITTI.

the sparsity gains are diminished. In our experiments, we use residual DCTs for video prediction, and absolute DCTs otherwise.

## 4  Experiments

To illustrate the flexibility and universality of our framework, and to test the performance of Transframer, we use a range of datasets and tasks, covering predictive video modelling, few-shot novel view synthesis, and a number of classic computer vision tasks. Our focus is on generality, and the ease of expressing a given task in the presented framework. Please see the appendix for model hyperparameters, training details, and additional ablation studies. A selection of example output videos can be seen at `https://sites.google.com/view/transframer`.

### 4.1  Video modelling

We train Transframer to predict the next frame given a series of input video frames. We supply one-hot time-steps as annotations as described in Table 1 (row 1), where time is specified relative to the target frame. For each video modelling task, we specify a minimum and maximum number of context frames. During training, for each element of a batch, we uniformly at random choose the number of context frames in the specified range, and pad the sequence to the maximum number of frames. In general, we choose the minimum number of frames to be equal to the number of context frames provided at test time.

**Metrics**  For video generation we use the following metrics: Structural Similarity Index Measure (SSIM, (Wang et al., 2004)), Peak Signal-to-noise Ratio (PSNR, (Huynh-Thu & Ghanbari, 2008)), Learned Perceptual Image Patch Similarity (LPIPS, (Zhang et al., 2018)) and Fréchet Video Distance (FVD, (Unterthiner et al., 2019)). FVD measures distributional similarity between sets of ground truth and generated videos, and is sensitive to image quality and temporal coherence. For RoboNet and KITTI we follow the

Table 2: Fréchet video distance for generated videos on BAIR and Kinetics600 datasets. For BAIR, we use a single context frame and generate 15 frames. For Kinetics600, we use 5 context frames and generate 11 frames.

| BAIR (64x64) | FVD↓ |
|---|---|
| DVD-GAN-FP (Clark et al., 2019) | 109.8 |
| VideoGPT (Yan et al., 2021) | 103.3 |
| TrIVD-GAN-FP (Luc et al., 2020) | 103.3 |
| Transframer (Ours) | 100.0 |
| Video Transformer (Weissenborn et al., 2020) | 94.0 |
| FitVid (Babaeizadeh et al., 2021) | **93.6** |
| Kinetics600 (64x64) | FVD↓ |
| Video Transformer (Weissenborn et al., 2020) | 170.0 |
| DVD-GAN-FP (Clark et al., 2019) | 69.1 |
| Video VQ-VAE (Walker et al., 2021) | 64.3 |
| TrIVD-GAN-FP (Luc et al., 2020) | 27.7 |
| Transframer (Ours) | **25.4** |

Table 3: Sample metrics for generated videos on KITTI and action-conditional RoboNet datasets. For KITTI, we use 5 context frames and generate 25 frames. For RoboNet, we use 2 context frames and generate 10 frames.

| | KITTI (64x64) | | | | RoboNet (64x64) | | | |
|---|---|---|---|---|---|---|---|---|
| | FVD↓ | PSNR↑ | SSIM↑ | LPIPS↓ | FVD↓ | PSNR↑ | SSIM↑ | LPIPS↓ |
| SVG (Villegas et al., 2019) | 1217.3 | 15.0 | 41.9 | 0.327 | 123.2 | 23.9 | 87.8 | 0.060 |
| GHVAE (Wu et al., 2021a) | 552.9 | 15.8 | 51.2 | 0.286 | 95.2 | 24.7 | 89.1 | 0.036 |
| FitVid (Babaeizadeh et al., 2021) | 884.5 | 17.1 | 49.1 | 0.217 | 62.5 | 28.2 | 89.3 | 0.024 |
| Transframer (Ours) | **260.4** | **17.9** | **54.0** | **0.112** | **53.0** | **31.3** | **94.1** | **0.013** |

literature and report the best SSIM, PSNR, and LPIPS scores over 100 trials for each video. For consistency with previous work we only report test-set FVD for Kinetics600 and BAIR.

**Datasets** We first evaluate our model on **BAIR** (Ebert et al., 2017), which is one of the most well studied video modelling datasets, consisting of short video clips of a single robot arm interactions in the unconditional setting. Table 2 shows the FVD score obtained by our model. While our model improves on some strong baselines such as TrIVD-GAN, it is out-performed by FitVid and Video Transformer. However, BAIR's small test set consisting of only 256 samples makes precise comparisons challenging, as discussed by Luc et al. (Luc et al., 2020). Moreover, both methods under-perform Transframer in all other benchmarks, with larger test sets.

**Kinetics600** (Carreira et al., 2018a) is an action recognition dataset, consisting of video clips of dynamic human actions across 600 activities, such as sailing, chopping, and dancing. It is a challenging dataset for video modelling, as the scenes feature complex motion, as well as sudden scene transitions. We use the same experimental set-up as DVD-GAN (Clark et al., 2019), operating at $64 \times 64$ resolution, and generate 11 frames given 5 context frames. We report test-set FVD, for 50k frames, sampling the context frames uniformly at random from the test-set clips. Our model obtains an FVD of 25.4, improving over the previous state-of-the-art model TrIVD-GAN-FP (Luc et al., 2020).

The **KITTI** dataset (Geiger et al., 2012) contains long video clips of roads and surrounding areas taken from the viewpoint of a car driver. It is less challenging than Kinetics600 due to smoother camera motion, and a more constrained visual setting (roads, cars, etc.). However it is a relatively small dataset, which makes generalization challenging. As such, we employ the augmentation methods of Babaeizadeh et al.

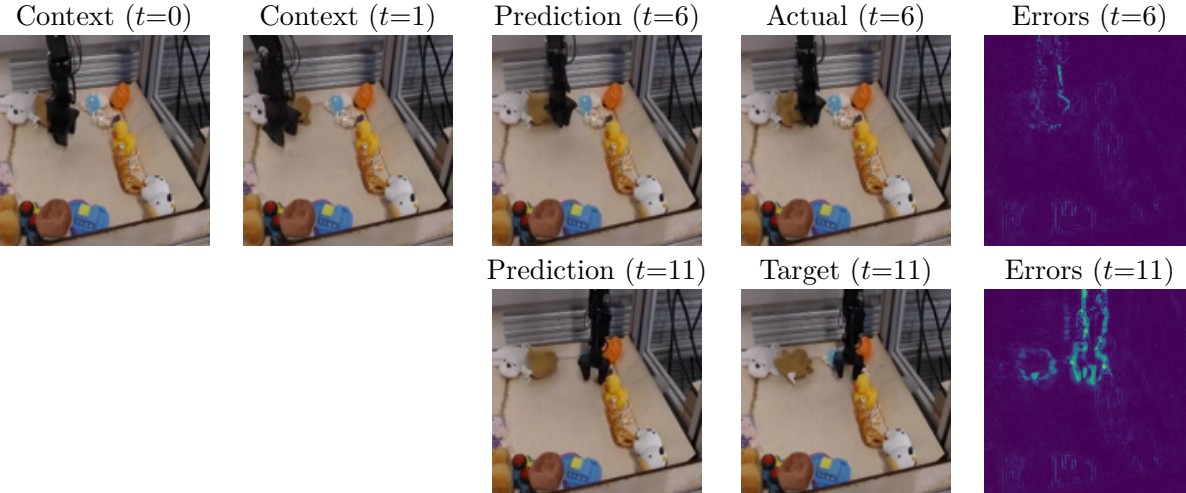

Figure 4: Action-conditional video generation on the RoboNet test set at 128x128 resolution. The action sequence for the output frames is provided to the model, and the frames are autoregressively generated, with a context length of up to 4.

(Babaeizadeh et al., 2021) to reduce overfitting. We follow the evaluation regime of Villegas et al. (Villegas et al., 2017), operating at 64×64 resolution, and generate 25 frames given 5 context frames, with test clips taken from 3 longer test clips at 5-frame strides. Table 3 shows that our model's performance improves on comparable methods on all metrics, with the improvement in LPIPS and FVD being the most substantial.

To evaluate our model in the action-conditional setting, we use **RoboNet** (Dasari et al., 2019), which consists of short video clips of robot arms interacting with objects, and provides robot action annotations as 5-dimensional vectors. The actions relate to the robot end-effector and gripper joint. We train at 64x64 and 128x128 resolutions, and evaluate using 2 context frames and 10 sampled frames on the test set specified by FitVid (Babaeizadeh et al., 2021). To process the input actions, we linearly project the 5-dimensional action-vectors and add them to input DCT image embeddings in the encoder. Table 3 shows the model's performance in comparison to existing methods, and Figure 4 shows an example of our model's outputs at 128x128 resolution. At 64x64 resolution, our model improves on alternatives in every metric by a large margin. At 128x128, we could not find comparable previous work, so we report our results in Appendix E for future comparisons.

**Long range video generation** As an additional proof-of-concept, we push the temporal limits of our model and generate 30 second videos at 25 frames per second (fps) conditioned on a single input image. At this frame rate, videos consist of 750 frames total. Generating these frames sequentially is challenging for a number of reasons. First, our architecture would need to process a large number of frames to model long-range dependencies; scaling such a model would be computationally expensive. Second, simply unrolling a conventional model with only a few frames of context often leads to globally incoherent generations or collapse of the output as generation errors compound over time-steps.

We address these issues with a two-stage procedure, where we first generate low fps videos using one model, and then interpolate to obtain the desired frame rate with another. We train the low fps model 1 fps videos with up to 15 context frames. At evaluation time we condition on a single context image, and sequentially predict 29 frames to yield a 1 fps video that is 30 seconds long in real time. We train a separate model using the same architecture to interpolate between consecutive 1 fps frames at 25 fps. During training, the interpolation model predicts a target frame conditioned on the penultimate frame as well as a single future frame. At test time the model takes consecutive 1fps frames as context, then generates the 25 fps frames successively. The first two rows in Table 1 describe both stages, and Figure 5 shows some example outputs.

| Context | 1s | 2s | 4s | 8s | 16s | 30s |
|---------|-----|-----|-----|-----|------|------|

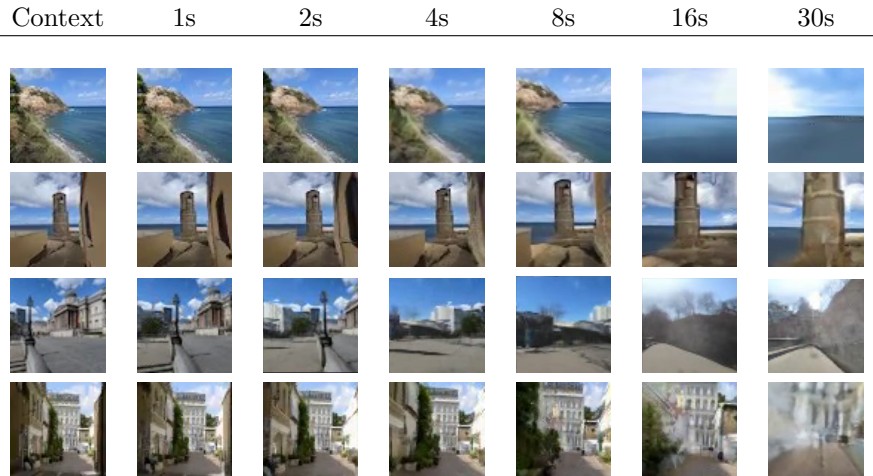

Figure 5: Given a single context frame we generate 30 seconds of video at 25fps. Full videos (including interpolated frames) and attributions for the context images are available at https://sites.google.com/view/transframer

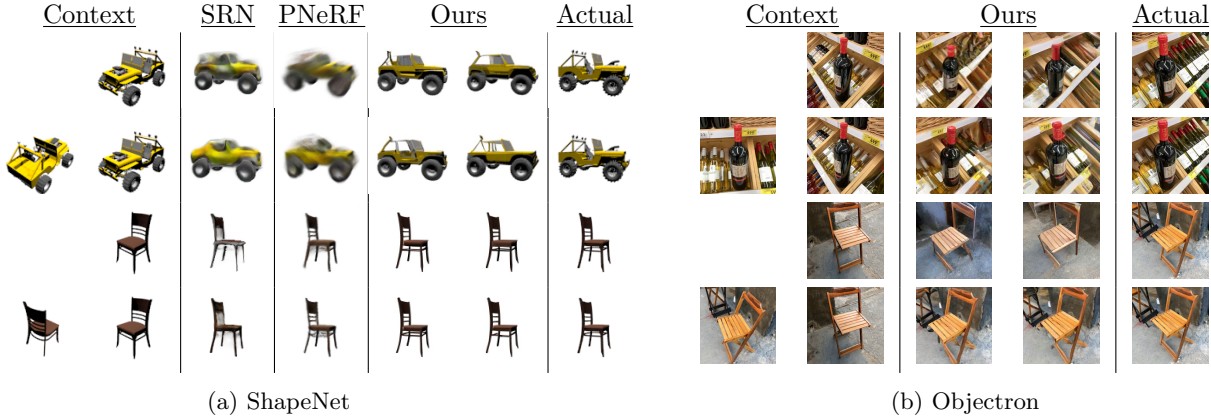

(a) ShapeNet                                    (b) Objectron

Figure 6: View synthesis on ShapeNet (a) and Objectron (b). For ShapeNet we also show the corresponding outputs from SRN (Vincent Sitzmann, 2019) and PixelNeRF (Yu et al., 2021).

**Impact of residual DCT representations** As described in Section 3.3, for video datasets where consecutive frames have a high level of temporal redundancy, images can be more compactly represented using the difference between DCT values in adjacent frames. We compare the data representations in Figure 3 with training curves for residual and absolute DCT sequences for BAIR and KITTI. Our experiments on both datasets suggest that residual representations provide an advantage, which is greater on BAIR, likely due to the static backgrounds and associated sparsity.

### 4.2   Novel view synthesis

We train Transframer for novel view synthesis by conditioning on the target camera position, as well as collection of annotated input views. We supply camera views as context and target annotations as described in Table 1 (row 3), and sample uniformly a number of context views, up to a specified maximum.

**Metrics** We follow Yu et al. (Yu et al., 2021) in using PSNR and SSIM to evaluate our model's few-shot view synthesis performance. These metrics are informative, but problematic in the case where the target view is not well determined by the input views. In these cases, the single prediction that optimizes PSNR will be a blurred average over possible target views. This penalizes models such as ours that make probabilistic

Table 4: View synthesis results on ShapeNet.

| | Chair (1 view) | | | | Chair (2 view) | | | |
|---|---|---|---|---|---|---|---|---|
| | FID↓ | PSNR↑ | SSIM↑ | LPIPS↓ | FID↓ | PSNR↑ | SSIM↑ | LPIPS↓ |
| PixelNeRF | 132.86 | **23.72** | **0.91** | 0.13 | 101.22 | **26.20** | **0.94** | 0.08 |
| Transframer (Ours) | **51.09** | 22.85 | **0.91** | **0.07** | **46.17** | 23.77 | 0.92 | **0.06** |
| | Car (1 view) | | | | Car (2 view) | | | |
| | FID↓ | PSNR↑ | SSIM↑ | LPIPS↓ | FID↓ | PSNR↑ | SSIM↑ | LPIPS↓ |
| SRN | 142.85 | 22.25 | 0.89 | 0.13 | 133.42 | 24.84 | 0.92 | 0.11 |
| PixelNeRF | 161.51 | **23.17** | **0.90** | 0.15 | 129.65 | **25.66** | **0.94** | 0.10 |
| Transframer (Ours) | **74.94** | 21.94 | 0.89 | **0.10** | **62.70** | 23.34 | 0.91 | **0.08** |

predictions, rather than a single point estimate. Hence, we introduce LPIPS and Fréchet Inception Distance (FID, (Heusel et al., 2017)) as additional metrics, both of which correlate with human perceptual preferences to a greater degree than the alternatives. We calculate per-scene FID between two sets: The ground truth set of the combined input and target views, and the sampled set consisting of all generated views combined with the input views.

**Datasets** We evaluate our model on **ShapeNet** benchmarks, in particular the chair and car subsets used by Yu et al. (Yu et al., 2021). The dataset consists of renders of 3D objects from the ShapeNet database. We train a single model across both classes, and don't provide class labels as input. As in Yu et al. (Yu et al., 2021), we evaluate using either 1 or 2 context views, and predict the remaining views in a 251-frame test-set. To process the input views we flatten and linearly project the 9-dimensional camera matrix and add them to input DCT image embeddings in the encoder. Table 4 compares our model to alternative approaches, and Figure 6 shows example results. Our model achieves worse PSNR and SSIM scores than baseline methods, but performs better on LPIPS and FID. As mentioned earlier, our model outputs samples, rather than a point estimate. As such it is unsurprising that it is outperformed on PSNR and SSIM by methods that output point estimates. Qualitatively, we observe that our model produces sharper outputs that capture the input views' textures, compared to somewhat blurry outputs for unseen views for PixelNeRF and SRN.

As a more realistic view-synthesis task, we train and evaluate on the **Objectron** dataset. Objectron consists of short, object-centered clips, with full object and camera pose annotations. The dataset features a diverse array of everyday objects, such as bottles, chairs and bikes, recorded in a variety of settings. To the best of our knowledge, no prior works have trained view-synthesis models in the few-shot setting on this dataset. We compare the performance of our model for varying numbers of context views, and describe our experimental setup in detail in Appendices F and G for future reference. Figure 6 shows example outputs for two scenes. When given a single input view, the model produces coherent outputs, but misses some features such as the crossed chair legs. Given two views, these ambiguities are resolved to a large extent.

### 4.3 Multi-task Vision

Different computer vision tasks are commonly handled using specialized (often intricately so) architectures and loss functions. In this section, we report an experiment where we jointly trained a single Transframer model using the same loss function on 8 different tasks and datasets: optical flow prediction from a single image, object classification, detection and segmentation, semantic segmentation (on two datasets), future frame prediction and depth estimation. We reduce all tasks to the same next-frame prediction task and do not adapt losses or architectural components. For the classification task, we use the ImageNet dataset (Deng et al., 2009) and consider predicting *one-hot images*, where each class is encoded as a 8x8 white patch on a black background. As image annotations we use a one-hot vector that determines the desired type of outputs, e.g., depth or optical flow. In all cases we used 256x256 resolution frames.

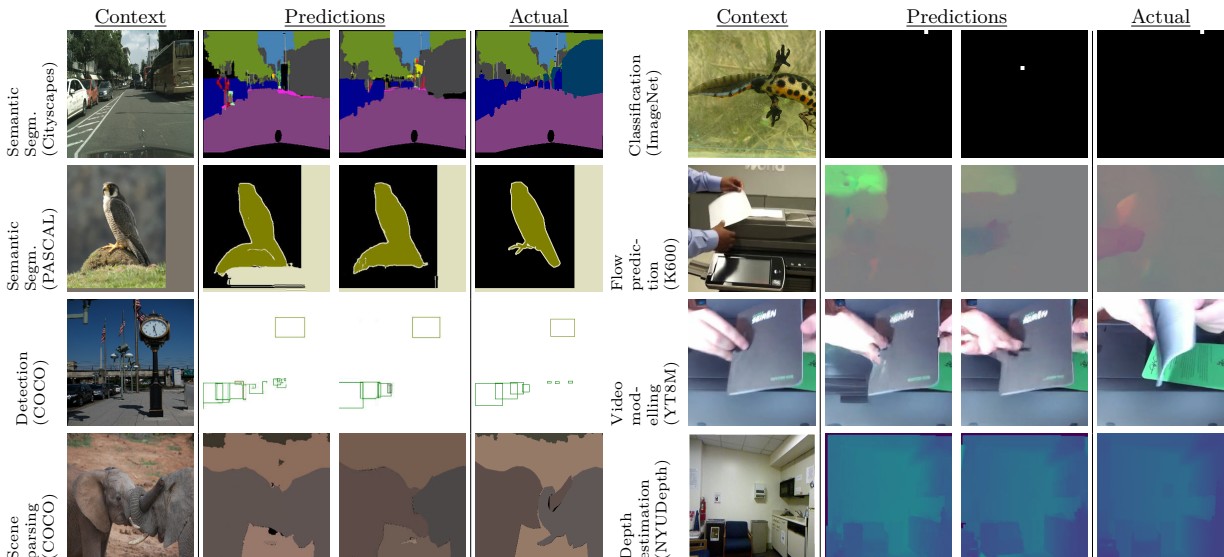

Figure 7: Example samples on the validation set of our multi-task setup, obtained using a single model. The model has no task specific parameters; in all cases it makes predictions conditioned on a context frame and a task embedding vector.

We trained a model using batches composed of randomly sampled elements from each dataset, rebalancing them slightly to accommodate their varied sizes. Several qualitative samples from the validation set are shown in Figure 7. It shows that the model learns to generate diverse samples across completely different tasks. On some tasks, such as Cityscapes, the model produces qualitatively good outputs, but model outputs on tasks like future frame prediction and bounding box detection are of variable quality, suggesting these are more challenging to model in this setting. Full experimental details can be found in Appendix H, along with quantitative performance metrics for the ImageNet classification and CityScapes semantic segmentation.

## 5 Conclusion

We introduced a general-purpose framework for conditional image prediction which unifies a range of image modelling and vision tasks. Our architecture, Transframer, models the conditional probability of image frames given target annotations and one or more context frames. Our results on video modelling and novel view synthesis confirm the quantitative strength of our approach, while our experiments applying the model to 8 standard vision tasks, including depth estimation, object detection, and semantic segmentation, demonstrate its broad applicability. These results show that general-purpose probabilistic image models can support challenging image and video modelling tasks, as well as multi-task computer vision problems traditionally addressed with task-specific methods.

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
