# OpenReview forum: "Transframer: Arbitrary Frame Prediction with Generative Models"
_TMLR — Accepted by TMLR_

### Review · Reviewer_1DLG · 2022-10-18

**Summary Of Contributions:**

This paper proposes the general framework Transframer for various vision tasks based on the DCTransformer.

Besides the concept of sparse DCT image representations, the suggested idea in this paper is simple: the authors try to model a variety of vision works into conditional image modeling tasks.

Specifically, Transframer is trained to predict the distribution of the target image, given the annotations of the target image and the context information (set of images and corresponding annotations).

In experiments, Transformer performs well on the video modeling and novel view synthesis task. Moreover, it shows its capability to handle various vision tasks through multi-task learning without any need for task-specific loss or architecture.

**Audience:**

Yes

**Broader Impact Concerns:**

Not applicable.

**Claims And Evidence:**

Yes

**Requested Changes:**

- Figure 2: What is z? Are they representations of each element from the previous block?
- Figure 3: In (a) Sparsity heatmaps, it is hard to know which image indicates videos from RoboNet or KITTI. In caption, (b) → (b, c).

**Strengths And Weaknesses:**

### Strengths

- Interesting idea of modeling video generation with a sparse image representation DCT and the autoregressive probabilistic model.
- Interesting extension of DCTrasnformer to video generation tasks.
- Outstanding performance compared to previous baselines on the video modeling and novel view synthesis tasks.
- Potential to be extended to solve various vision tasks with a unified framework and loss function.

### Weaknesses

Please consider that I may miss some critical flaws in the experimental setups since I am not an expert on video modeling and novel view synthesis tasks.

- Some of the writings are not clear. See the requested changes part for details.

---

### Review · Reviewer_L2ne · 2022-11-05

**Summary Of Contributions:**

This paper proposes Transframer, a unified probabilistic framework for conditional image prediction and generation. Specifically, Transframer utilizes a transformer and a U-Net to learn the conditional probability of the target given known context and target annotations. The proposed framework has been tested on many computer vision tasks including video modeling, novel view synthesis, semantic segmentation, image classification, and optical flow prediction.

**Audience:**

Yes

**Broader Impact Concerns:**

No concerns.

**Claims And Evidence:**

No

**Requested Changes:**

- Provide a clear explanation of the differences between the proposed method and existing methods, especially the loss functions (critical).

- Provide an explanation of how the proposed method provides uncertainty estimation and provide empirical demonstrations (critical).


**Strengths And Weaknesses:**

Strengths

- Providing a unified framework for different problems is an important question and will be of interest to the community.

- Combining a transformer with a U-Net to learn the conditional probabilities is a reasonable idea.

- Extensive empirical results are provided to demonstrate the proposed methods on a variety of computer vision tasks.

Weaknesses
- Can the authors be more specific about the difference between the proposed framework and the existing frameworks? As far as I understand, using a model to learn p(target| context) seems a standard way to make predictions on many computer vision tasks. The authors mentioned that the same framework has been used in novel view synthesis and video generation but not semantic segmentation, depth estimation, or optical flow estimation. It will be better to give more explanation on the differences. Since even task-specific loss functions are used, they might be equivalent to some conditional probability and thus still fit into the probabilistic framework. I think the authors may also have to use different loss functions for different tasks (though all of them can be interpreted as conditional probabilities).

- How does the proposed framework estimate uncertainty? The authors claimed that the proposed probabilistic framework can provide a distribution over predictions and thus offer uncertainty estimation. It is unclear what the outputs are from the proposed model and how it outputs a distribution instead of a point estimation. In all experiments, the authors do not show the uncertainty estimation results.

---

### Review · Reviewer_Kiok · 2022-12-14

**Summary Of Contributions:**

The paper introduces a framework to perform various computer vision tasks under the umbrella of conditional frame prediction using a generative model that operates on compressed data representations (DCT). The authors combine this with a new architecture named the Transframer to obtain competitive performance on multiple computer vision tasks including coherent video generation.

**Audience:**

Yes

**Claims And Evidence:**

Yes

**Requested Changes:**

I think most of the points listed in the cons above can be addressed to secure my recommendation

**Strengths And Weaknesses:**

**Pros:**
* The experiments are exhaustive and support the claims very well. The results are very competitive across various tasks and extremely impressive, especially in video modeling and novel view synthesis.
* The paper is very well structured; makes it easy to follow
* The core idea is simple yet effective
* The authors have provided training details that should help with reproducibility
* Long-range video generation using two-step generation+interpolation is smart

**Cons:**
* Most of the components used by the method exist in the literature which reduces the novelty. (This is not a criterion that would affect my decision, I am merely stating it as an observation)
* Section 3.2 can do with improved details. The authors just textually describe the model which makes it really hard to pass without multiple iterations, improved figures that show more details of the shared U-Net (NF block), and how the representations are attentions are calculated would be really helpful.
* Although the paper is well structured, equations (5, 6, 7) could do with improved symbols. it is quite difficult to understand what $\{I_k^{target} \}_k$ means without multiple iterations.
* Section 3.1 mentions framework allows for object detection and recognition but the authors have not provided any performance metrics (just visualizations in the appendix)
* Is there a specific reason why the sparse DCT list needs to be sorted? The operations look equivariant/invariant.
* The results in the appendix showing the results with respect to the DCT quality are nice. I am curious as to what happens when the DCT quality is different between training and testing. Does the model still generalize well?
* Some results on training/inference wall-clock time would be nice
* I could not find any code/ checkpoints for reproducibility

---

### Decision · Action_Editors · 2023-02-14

**Recommendation:** Accept as is

**Comment:**

The paper has been reviewed by four reviewers. Three recommended "leaning accept" and one recommended "accept".

The general consensus is that the experiments in the paper are thorough and the performances of the proposed method are impressive. The reviewers also find the proposed transframer architecture interesting.

The reviewers asked the authors to clarify the novelty compared to existing work, and suggested improvements on presentation. The rebuttal addresses these points.

Overall this is an interesting and useful paper. I recommend acceptance.

**Audience:**

The paper is interesting to researchers working on generative AI and computer vision.

**Claims And Evidence:**

This paper proposes a transframer framework for image frame prediction using U-net and transformer decoder. The proposed method is applied to a wide range of vision tasks, including image segmentation, novel view synthesis, and video interpolation.